# Evaluating immune response and metabolic related biomarkers pre-allogenic hematopoietic stem cell transplant in acute myeloid leukemia

Sharareh Siamakpour-Reihani[1], Felicia Cao[2], Jing Lyu[2], Yi Ren[3], Andrew B. Nixon[2], Jichun Xie[3], Amy T. Bush[1], Mark D. Starr[2], James R. Bain[2,4], Michael J. Muehlbauer[4], Olga Ilkayeva[4], Virginia Byers Kraus[2,4], Janet L. Huebner[4], Nelson J. Chao[1], Anthony D. Sung[1]*

1 Division of Hematologic Malignancies and Cellular Therapy, Duke University School of Medicine, Durham, North Carolina, United States of America, 2 Department of Medicine, Duke University School of Medicine, Durham, North Carolina, United States of America, 3 Department of Biostatistics and Bioinformatics, Duke University School of Medicine, Durham, North Carolina, United States of America, 4 Duke Molecular Physiology Institute, Duke University School of Medicine, Durham, North Carolina, United States of America

* anthony.sung@duke.edu

## Abstract

Although hematopoietic stem cell transplantation (HCT) is the only curative treatment for acute myeloid leukemia (AML), it is associated with significant treatment related morbidity and mortality. There is great need for predictive biomarkers associated with overall survival (OS) and clinical outcomes. We hypothesized that circulating metabolic, inflammatory, and immune molecules have potential as predictive biomarkers for AML patients who receive HCT treatment. This retrospective study was designed with an exploratory approach to comprehensively characterize immune, inflammatory, and metabolomic biomarkers. We identified patients with AML who underwent HCT and had existing baseline plasma samples. Using those samples (n = 34), we studied 65 blood based metabolomic and 61 immune/inflammatory related biomarkers, comparing patients with either long-term OS (≥ 3 years) or short-term OS (OS ≤ 1 years). We also compared the immune/inflammatory response and metabolomic biomarkers in younger vs. older AML patients (≤30 years vs. ≥ 55 years old). In addition, the biomarker profiles were analyzed for their association with clinical outcomes, namely OS, chronic graft versus host disease (cGVHD), acute graft versus host disease (aGVHD), infection and relapse. Several baseline biomarkers were elevated in older versus younger patients, and baseline levels were lower for three markers (IL13, SAA, CRP) in patients with OS ≥ 3 years. We also identified immune/inflammatory response markers associated with aGVHD (IL-9, Eotaxin-3), cGVHD (Flt-1), infection (D-dimer), or relapse (IL-17D, bFGF, Eotaxin-3). Evaluation of metabolic markers demonstrated higher baseline levels of medium- and long-chain acylcarnitines (AC) in older patients, association with aGVHD (lactate, long-chain AC), and cGVHD (medium-chain AC). These differentially expressed profiles merit further evaluation as predictive biomarkers.

**Data Availability Statement:** All relevant data are within the paper and its Supporting Information files.

**Funding:** 1. American Society of Hematology (ASH), (PI: Anthony D Sung) https://www.hematology.org/ 2. NIH/National Institute on Aging 1R21AG066388-01 award (PI: Anthony D Sung). https://reporter.nih.gov/search/5NyHmfz0skuWJC2JAuThRw/project-details/9980757 3. NIH/National Institute on Aging Duke Pepper Older Americans Independence Center P30 AG028716, (PI: Schmader, Mini #6, PI of Mini: Anthony D Sung) https://reporter.nih.gov/search/KsvVX_8rwUKk6CXVf2k0vg/project-details/9971412 The funder's role included all aspects of the study design, supervision of the data collection, data analysis, decision to publish, and preparation of the manuscript.

**Competing interests:** The authors have declared that no competing interests exist.

## Introduction

Acute myeloid leukemia (AML) is a molecularly and clinically heterogeneous disease with biological complexity [1]. There have been major advances in understanding the genetic factors related to AML and the disease biology and pathophysiology during the past thirty years. However, induction chemotherapy and consolidation therapy with allogenic hematopoietic stem cell transplant (HCT) or additional chemotherapy remains the standard treatment, with about 20–30% of AML patients never achieving remission [2]. This is true specifically for intermediate or high-risk AML patients who have increased risk of relapse, and HCT remains the best and only chance for cure. Yet, HCT is associated with severe treatment related morbidities such as infections and graft-versus-host disease (GVHD), and risk of non-relapse mortality ranges from 8–38% [3, 4]. Low rates of complete remission (CR) (30–50%) and poor overall survival (OS) (15–55% at one year) have been attributed to a variety of reasons including increased incidence of poor-risk cytogenetics, mutations such as FMS-like tyrosine kinase 3 with the internal tandem duplication (*FLT3*-ITD), increased activation of RAS, Src, and TNF pathways, and intrinsic resistance of leukemic blasts to therapeutic agents [5–7].

The incidence of AML increases with age, with the biology of AML changing with age [5]. Unfortunately, many older AML patients are considered unfit for intensive treatment because of frailty and the risk of fatal toxicity [1, 2, 8, 9]. Even in older patients who receive intensive treatment, outcomes remain unsatisfactory with low rates of CR, poor disease-free survival (DFS) and OS [3, 4]. Given improvements in therapeutic regimens and supportive care (including infection control and transfusion support), in patients younger than 60, AML is now cured in approximately 35–40% of cases. However, for AML patients >60 years, although the prognosis has improved, survival is still poor, with OS<1 year compared to OS of almost 3 years for patients aged 15 to 55 [5, 8–10].

Genomic profiles and multiple somatically-acquired mutations can be used for AML characterization, affecting prognosis and serving as predictive biomarkers. Genetic alterations in AML can be divided into three groups: 1) cytogenic abnormalities such as translocations, inversions, deletions, trisomies and monosomies, 2) cytogenetically normal but with gene mutations, such as in NPM1, FLT3, CEPBA, RAS, WT1, and TP53 and 3) epigenetic mutations, such as DNMT3A, IDH1/2, and TET (C). In AML patients, genetic screening is used for prognostic categorization (favorable, intermediate, and poor risk) and the subsequent selection of treatment strategies. Currently, the World Health Organization (WHO) classification of myeloid neoplasms distinguishes between AML with mutations in RUNX1 and AML with the BCR-ABL1 fusion. In addition, the 2017 European LeukemiaNet recommendations for AML adds mutations in three genes—RUNX1, ASXL1, and TP53—for risk stratification of AML [11, 12]. The random accumulation of mutations due to aging is one reason that AML is considered a disease of the elderly. Despite these genetic associations, there is a need for additional blood based markers for phenotyping patients because genetics alone are not fully prognostic.

Aging is a complex process that is characterized by physical, molecular, and deleterious immune and metabolic changes [13, 14]. Biological aging is characterized by dysregulated immune and metabolic homeostasis [15]. Regardless of the cause, a common feature of aging and many age-related diseases is chronic inflammation in the absence of infection (termed "inflammaging"). Changes in circulating levels of blood based biomarkers of inflammation/immune response have been shown to be associated with inflammaging. Examples of such biomarkers are C-reactive protein (CRP), interleukin-6 (IL-6), tumor necrosis factor alpha (TNFa) and its soluble receptors (tumor necrosis factor receptors I (TNFR-I) and II (TNFR-II), vascular cell adhesion molecule I (VCAM-I), and D-dimer [14, 16, 17]. The Pepper Panel, developed by the Duke Pepper Center for Aging, includes biomarkers of aging,

inflammation, mitochondrial dysfunction, and dysregulated protein metabolism. Age was positively correlated with TNF-α, TNFR-I, TNFR-II, IL-6, IL-2, VCAM-1, D-dimer, matrix metalloproteinase-3 (MMP-3) and adiponectin markers; age was negatively correlated with G-CSF, RANTES, and paraoxonase activity [14].

Metabolic changes are part of the aging process. Aging related dysregulation of inflammatory/immune responses occurs in tandem with metabolomic dysregulation. However, the mechanisms are poorly understood [14]. It has been reported that changes in such circulating, energy–related metabolites as acylcarnitines, carbohydrates, and amino acids (AA), can be associated with age, BMI and insulin resistance. Metabolic markers such as adiponectin, glycine, nonessential AA, and relative proportions of circulating large, neutral AAs (LNAAs) and medium-chain acylcarnitines have been suggested as markers of metabolic health. For example, higher plasma concentrations of glycine have been reported to be associated with better metabolic health [14, 18–20].

We hypothesized that circulating metabolic, inflammatory, and immune molecules have potential as predictive biomarkers for AML patients who receive HCT treatment. We have studied those biomarkers in AML-HCT patients who have shorter vs longer OS (OS of less than one year (OS ≤1) or more than three years (OS ≥3). We compared the blood based biomarkers and metabolomics profiles in younger vs. older AML patients (≤30 years vs. patients ≥ 55 years). We also analyzed the biomarker and metabolomics profiles for their association with clinical outcomes, namely OS, chronic graft versus host disease (cGVHD), acute graft versus host disease (aGVHD), infection and relapse.

## Methods

### Patient population

We retrospectively identified patients with AML who underwent HCT, and had baseline (pre-HCT) EDTA plasma samples. The samples were stored according to our Duke Health Institutional Review Board (IRB)-approved protocols (IRB# Pro00006268 and Pro00100650). Our protocols involved obtaining witnessed informed consent for sample collection for future research purposes, as well as use of associated clinical data.

Fully anonymized samples were thawed, aliquoted, refrozen and stored at −80˚C until tested. All samples underwent one freeze-thaw cycle before ELISA analysis. In order to compare biomarkers in older vs. younger patients and those with good vs. poor OS outcomes, we selected samples based on patient age (≤30 years vs. ≥ 55 years old) and OS outcomes (OS ≤ 1 years vs. OS≥ 3 years). (**Table 1**). The age cut-offs were arbitrary selection and mostly based on what we had samples for. Though there is support that AML in patients >55years behaves differently than AML in patients <55 years [5].

### Biomarkers of inflammation and aging

To evaluate plasma biomarkers of inflammation/immune response, we employed the Meso Scale Quickplex SQ 120 system from Meso Scale Diagnostic (MSD), LLC. (Rockville, MD). We used the V-PLEX Human Biomarker 54-Plex Multiplex Plates, (Cat#K15248D, MesoScale Diagnostics, Rockville, MD). The 54-Plex is designed to provide a multiplex assay for measuring markers involved in inflammation response and immune system regulation [21–23]. As previously mentioned the biological aging is characterized by dysregulated immune and metabolic homeostasis [15]. We chose the 54–plex to get a broad understanding of the potential differences in the immune/inflammatory response biomarker based on the patients' age, survival and their association with clinical outcomes. The 54-Plex assay evaluated the following markers: CRP, eotaxin, eotaxin-3, FGF (basic), GM-CSF, ICAM-1, IFN-γ, IL-10, IL-12/IL-23p40,

**Table 1. Demographic data and patient characteristics.**

| | All Patients | Older than 55 | Younger Than 30 | |
|---|---|---|---|---|
| | N = 34 (100%) | N = 18 (52.9%) | N = 16 (47.1%) | P-Value |
| **Age** | | | | |
| Median (IQR) | 56.5 (21–59) | 59 (57–65) | 21 (20–23.5) | < .0001 |
| **Sex** | | | | |
| F | 19 (50%) | 10 (55.6%) | 9 (56.3%) | 0.9675 |
| M | 15 (39.5%) | 8 (44.4%) | 7 (43.8%) | |
| **Race** | | | | |
| Black | 6 (15.8%) | 1 (5.6%) | 5 (31.3%) | 0.0678 |
| Pacific Islander | 1 (2.6%) | 0 (0%) | 1 (6.3%) | |
| White | 27 (71.1%) | 17 (94.4%) | 10 (62.5%) | |
| **Conditioning** | | | | |
| Myeloablative | 21 (55.3%) | 8 (44.4%) | 13 (81.3%) | 0.0275 |
| Non-myeloablative | 17 (44.7%) | 10 (55.6%) | 3 (18.8%) | |
| **Graft Source** | | | | |
| Bone marrow | 1 (2.6%) | 1 (5.6%) | 0 (0%) | 0.4388 |
| Cord | 14 (36.8%) | 6 (33.3%) | 8 (50%) | |
| Peripheral blood progenitor cells (PBPCs) | 19 (50%) | 11 (61.1%) | 8 (50%) | |
| **Donor Type** | | | | |
| Related | 12 (31.6%) | 7 (38.9%) | 5 (31.3%) | 0.6418 |
| Unrelated | 22 (57.9%) | 11 (61.1%) | 11 (68.8%) | |
| **Survival group** | | | | |
| OS<1yr | 28 (73.7%) | 13 (72.2%) | 11 (68.8%) | 0.8245 |
| OS>3yrs | 10 (26.3%) | 5 (27.8%) | 5 (31.3%) | |
| **aGvHD** | | | | |
| No | 20 (52.6%) | 11 (61.1%) | 8 (50%) | 0.5149 |
| Yes | 18 (47.4%) | 7 (38.9%) | 8 (50%) | |
| **cGvHD** | | | | |
| No | 26 (68.4%) | 12 (66.7%) | 14 (87.5%) | 0.1529 |
| Yes | 12 (31.6%) | 6 (33.3%) | 2 (12.5%) | |
| **Relapse** | | | | |
| No | 30 (78.9%) | 15 (83.3%) | 11 (68.8%) | 0.3170 |
| Yes | 8 (21.1%) | 3 (16.7%) | 5 (31.3%) | |
| **infection** | | | | |
| No | 8 (21.1%) | 3 (16.7%) | 4 (25%) | 0.5486 |
| Yes | 30 (78.9%) | 15 (83.3%) | 12 (75%) | |

IL-12p70, IL-13, IL-15, IL-16, IL-17A, IL-17A/F, IL-17B, IL-17C, IL-17D, IL-1RA, IL-1α, IL-1β, IL-2, IL-21, IL-22, IL-23, IL-27, IL-3, IL-31, IL-4, IL-5, IL-6, IL-7, IL-8, IL-8 (HA), IL-9, IP-10, MCP-1, MCP-4, MDC, MIP-1α, MIP-1β, MIP-3α, PlGF, SAA, TARC, Tie-2, TNF-α, TNF-β, TSLP, VCAM-1, VEGF-A, VEGF-C, VEGF-D, and VEGFR-1/Flt-1. Each individual multiplex panel was run at the prespecified dilution for optimal performance and all samples were tested in duplicate.

To evaluate plasma biomarkers of aging, we also used markers from the Duke Pepper Panel [14], and other markers reported as significant in GVHD (REG3 and ST2) [24, 25]. The Duke Pepper Panel included the following 12 blood based markers: adiponectin, IL-2, IL-6, TNF- α, TNFRI*, TNFRII*, D-dimer*, G-CSF, regulated on activation, normal T cell expressed and secreted (RANTES)*, MMP-3*, paraoxonase*, VCAM-1 [14]. Many of these biomarkers were

already included in the 54-Plex panel, markers denoted by * were analyzed using separate enzyme-linked immunosorbent assay (ELISAs) including: RANTES (MesoScaleDiscovery Cat#F21ZN-3), TNFRI/TNFRII/MMP-3 (MesoScaleDiscovery Cat#F210V-3/F21ZS-3) D-dimer (Sekisui Diagnostics Cat#602), IL6Ra (R&D Systems Cat#DR600), and Paraoxonase (Invitrogen Molecular Probes Cat#E33702).

To evaluate plasma biomarkers of GVHD, in addition to IL6R as described above, samples were analyzed using ELISAs for regenerating family member 3 alpha (REG3A) (MBL Cat#5323/5310) [26–28]: REG3A is an anti-microbial peptides (AMP) and has been identified and validated as a diagnostic biomarker of gastrointestinal (GI) GVHD [24, 29]. Plasma REG3A plasma concentrations have been reported to be higher in GI GVHD patients [24, 30]. We have also evaluated suppression of tumorigenicity 2 (ST2, also known as interleukin 1 receptor like 1, (IL1RL1), IL1RL1/ST2 MesoScaleDiscovery Cat#F214H-3). ST2 has been evaluated as biomarker of GVHD, with elevated ST2 levels being associated with therapy-resistant GVHD and mortality [25, 30].

## Biomarkers of metabolism

We evaluated 65 metabolic biomarkers including: amino acids (N = 15), acylcarnitines (N = 45), and conventional clinical analytes (N = 5). Amino acids and acylcarnitines were analyzed by flow injection electrospray-ionization tandem mass spectrometry and quantified by isotope or pseudo-isotope dilution using methods described previously [31, 32]. Conventional analytes, including non-esterified fatty acids (NEFA), triglycerides, glycerol, 3-hydroxybutyrate, and lactate and were measured using a Beckman DxC 600 clinical analyzer (Brea, CA). Reagents for 3-hydroxybutyrate and NEFA were from Wako (Mountain View, CA), and those for lactate and triglycerides were from Beckman (Brea, CA). We measured free glycerol using the initial absorbance in the triglycerides assay, a signal that is normally blanked in the test procedure preceding the addition of lipase.

## Statistical analysis for biomarker studies

The 54-plex immune/inflammatory response biomarkers, the additional plasma biomarkers, and metabolomic biomarkers were analyzed using Cox Proportional Hazard models with OS as the response, and age groups, baseline biomarkers, and their interactions as predictors. Corresponding to the Cox PH model, p-values were reported from Chi-square test with df = 2 [33]. Box plots were used to illustrate the variability of the markers for association with clinical outcomes. Boxes represent 25th (Q1) and 75th (Q3) percentiles; Horizontal lines indicate the medians; Upper whiskers indicate maximum; Lower whiskers indicate minimum; Points indicate any observations outside the whiskers. Expression levels were log2-transformed and analyzed as continuous measures (Please see S1 File for the data underlying the findings). OS groups with survival of equal and less than one year or more than three years (OS ≤ 1 years or OS ≥ 3 years) were used as the output in logistic regression models. OS time was defined as time from transplant to death/last follow-up. Baseline samples' biomarker levels in different age and OS groups were analyzed using a Wilcoxon rank-sum test [34]. Effect sizes of each biomarker for aGVHD, cGVHD, relapse and infection were assessed using logistic regression models. Odds ratios (OR), score test p-values, and 95% CI were reported for the logistic regression models assessing biomarker effects on clinical outcomes (aGVHD, cGVHD, relapse and infection). Multiple comparison was addressed within a framework of control of False Discovery Rate (FDR) using the method by Storey [35, 36]. However, due to the limited sample size and number of markers analyzed, we used 0.1 as the significant q-value cutoff.

All analyses were performed using the R Statistical Environment [R], [6] and extension packages from CRAN [37, 38]. The analyses were conducted with adherence to the principles of reproducible analysis using the knitr package for generation of dynamic reports [39].

## Results

### Blood based biomarkers

Evaluation of the blood based immune/inflammatory response markers in younger vs. older HCT patients revealed significant differences in the pre-HCT baseline levels of the biomarkers (Table 2 and Fig 1). Compared to younger patients, older patients tended to have higher values of the following markers (Fig 1A–1H): interleukin-6 (**IL6** p-value = 0.002), interleukin-12 (**IL12/ IL12p70** p-value = 0.031), interleukin-16 (**IL16** p-value = 0.022), interleukin-17 (**IL17B** p-value = 0.028, **IL17C** p-value = 0.007, **IL17D** p-value = 0.032, **IL17A** p-value = 0.036), interleukin-27 (**IL27** p-value = 0.008), interleukin-1 receptor antagonist (**IL1RA** p-value = 0.015), macrophage inflammatory protein (**MIP1a** p-value = 0.041), placental growth factor (**PlGF** p-value = 0.001), thymic stromal lymphopoietin (**TSLP,** p-value = $1.0 \times 10^{-4}$), tumor necrosis factor alpha (**TNF-α** (also known as TNF) p value = 0.010), cell receptors tumor necrosis factor receptor I (**TNFRI**, p-value = 0.013) and tumor necrosis factor receptor II (**TNFRII,** p-value = 0.005), cellular adhesion protein vascular cell adhesion molecule 1 (**VCAM1** p-value = 0.0002), and apolipoprotein serum amyloid A (**SAA**, p-value = 0.019) (Table 2). This higher baseline inflammatory state may predispose older patients to worse outcomes after HCT, including the potential for increased risk of GVHD [40, 41]. Our data demonstrates that only interleukin-23 (**IL23**[*], p-value = 0.022) was significantly higher in younger patients (age ≤30 years) compared to older patients (age ≥55 years) (Fig 1I).

**Table 2. Baseline marker levels comparing younger (age ≤30 years) to older patients (age ≥55 years).**

| Marker Base level | p value |
|---|---|
| TSLP | 0.00009 |
| VCAM1 | 0.0002 |
| PlGF | 0.001 |
| IL6 | 0.002 |
| IL17C | 0.007 |
| IL27 | 0.008 |
| TNFRII[*] | 0.005 |
| TNF-α | 0.010 |
| TNFRI[*] | 0.013 |
| IL1RA | 0.015 |
| SAA | 0.019 |
| IL23 [**] | 0.022 |
| IL16 | 0.022 |
| IL17B | 0.028 |
| IL12p70 | 0.031 |
| IL17D | 0.032 |
| IL17A | 0.036 |
| MIP1a | 0.041 |

[*] are separate ELISAs not part of the 54-plex.

[**] Higher baseline levels in younger patients.

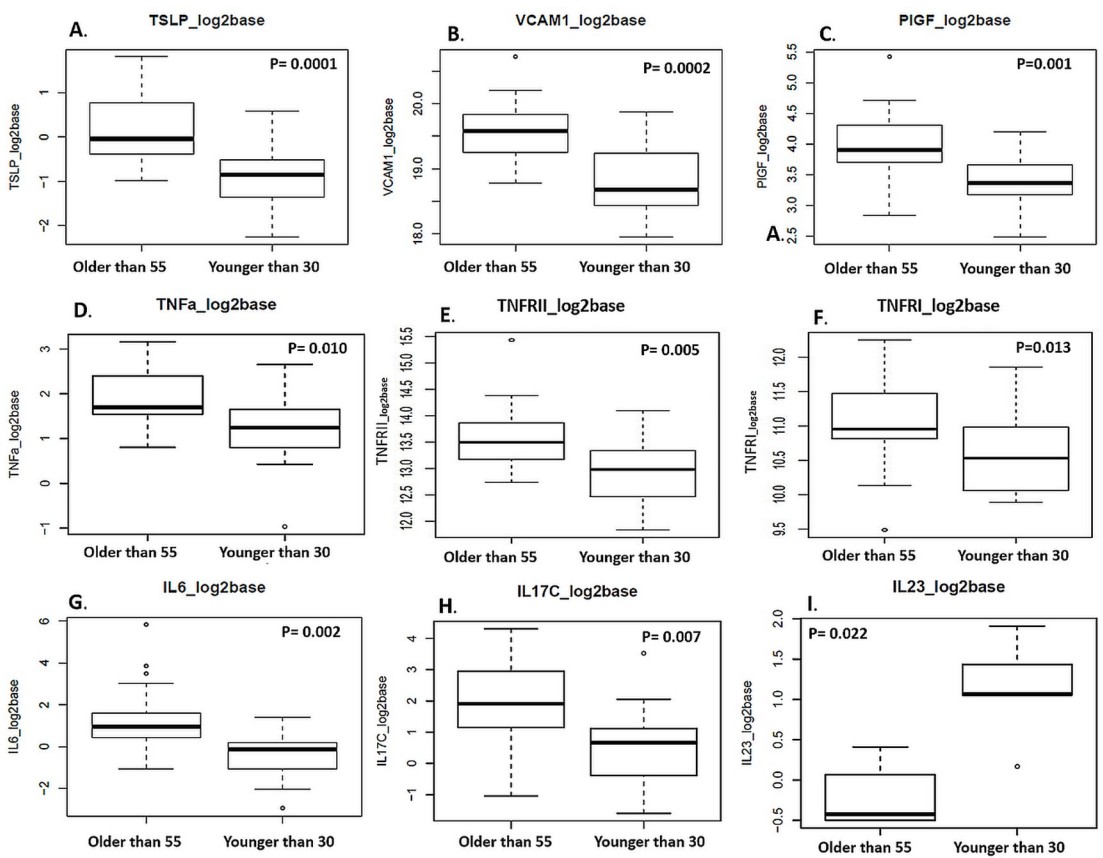

**Fig 1. Evaluating baseline levels of blood based biomarkers in younger vs. older HCT patients (≤30 years vs. ≥ 55 years old).**
The Wilcoxon rank-sum test was used to compare biomarker levels of baseline in different age groups (≤30 years vs. ≥ 55 years old). Data is presented as a boxplot. Of the n = 18 blood- based biomarkers with significant p-value, only IL23 was significantly higher in younger patients (age <30 years) compared to older patients (age >55 years). The rest of the markers had higher baseline levels in older patients (for the full list of the 18 markers please refer to Table 2).

**Association between OS and baseline biomarker expression.** Evaluating the association between OS and baseline biomarker expression using the Wilcoxon Rank-Sum test revealed that three markers C-reactive protein (**CRP**, p-value = 0.027), **SAA** (p-value = 0.018) and **IL13**, p-value = 0.017) had significant association with survival regardless of age. Baseline levels are lower in all the three markers in patients with OS≥ 3 years (Fig 2).

**Association between baseline biomarker levels and clinical outcomes (aGVHD, cGVHD, relapse and infection).** Baseline levels of three biomarkers were significantly associated with post-HCT "**relapse**". Patients with post HCT relapse had lower baseline levels of Interleukin 17D (**IL17D**, OR = 0.17, 95% CI = (0.03, 0.63), p-value = 0.020) and fibroblast growth factor 2 (**bFGF** (also known as (FGF2), OR = 0.63, 95% CI = (0.38, 0.94), p-value = 0.038). In contrast, patients with relapse had higher baseline levels of **eotaxin-3** (OR = 3.89, 95% CI = (1.43, 15.64), p-value = 0.027) (Fig 3A–3C).

Only **D-dimer** was significantly associated with **infection** post-HCT; patients who developed infection post-HCT had higher baseline levels of D-dimer (OR = 2.68, 95% CI = (1.18, 7.91), p-value = 0.038) (Fig 3D).

One baseline plasma marker was significantly associated with **cGVHD**: fms-like tyrosine kinase 1 (**Flt-1**, also known as **VEGFR-1**, OR = 1.71, 95% CI (1.13, 2.78), p-value = 0.017). Patients who developed cGVHD post-HCT had higher baseline expression levels of Flt-1 (Fig 3E).

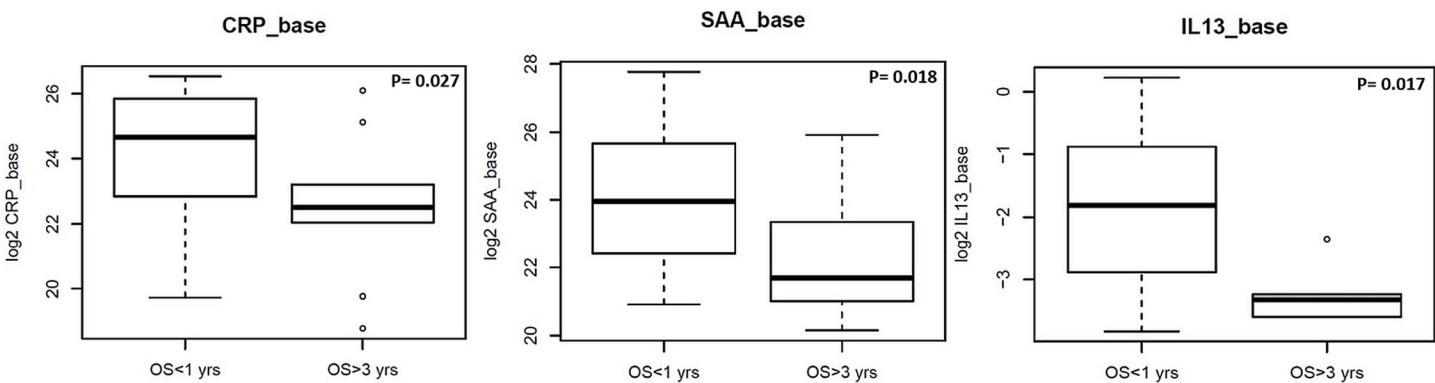

**Fig 2. Association between OS and baseline biomarker.** We have evaluated the association between components of the blood based biomarkers' baseline levels and OS. Data is presented as a boxplot. OS groups (OS ≤ 1 years or OS ≥ 3 years) were used as the output in logistic regression models. Logistic regression with OS groups as response, age groups, marker and their interaction as predictors was performed. The function was used to provide p-values of testing whether marker has significant effect on OS groups in either age group. Three markers (CRP, SAA and IL13) showed significant association with survival. Baseline levels are lower in all the three markers in patients with longer OS (OS≥ 3) years.

Two baseline plasma markers were significantly associated with **aGVHD**: interleukin 9 (**IL-9**, OR = 0.37, 95% CI (0.16, 0.71), p-value = 0.009); and **eotaxin-3** (also known as C-C motif chemokine ligand 26 (**CCL26**), OR = 0.35, 95% CI = (0.11, 0.84), p-value = 0.040). Patients who developed aGVHD post-HCT had lower baseline expression levels of both markers (Fig 3F and 3G).

## Metabolic biomarkers

The aging process is a complex, characterized by physical, molecular, immune, and metabolic changes that can cause functional decline [24, 28]. In addition to evaluating plasma biomarkers of immune and inflammatory response, we have **65** metabolites, including: amino acids (N = 15), acylcarnitines (N = 45), and conventional clinical analytes (N = 5). Markers were evaluated for association with age, OS and clinical outcomes (aGVHD, cGVHD, relapse and infection).

**Association between baseline metabolic biomarkers and clinical outcomes (aGVHD, cGVHD, relapse and infection).**   No metabolic marker showed significant association with survival for AML/HCT patients.

Baseline circulating **lactate** was associated with development of **aGVHD** post HCT. Patients with aGVHD had lower baseline levels of lactate (LACT, OR = 0.24, 95% CI = (0.06, 0.69), p-value = 0.019) (Fig 3H). Baseline **acylcarnitines (AC)** also demonstrated association with **aGVHD,** specifically C2-acylcarnitine, 3-hydroxy-tetradecanoyl carnitine or dodecanedioyl carnitine (**C14-OH/C12-DC**, OR = 0.24, 95% CI = (0.05,0.75), p-value = 0.028), 3-hydroxy-tetradecenoyl carnitine (**C14:1-OH**, OR = 0.35, 95% CI = (0.11, 0.93), p-value = 0.049), and long-chain acylcarnitines. Patients with aGVHD had lower baseline levels of C14-OH/C12-DC and C14:1-OH compared to patients with no aGVHD. Baseline levels of a medium-chain AC markers of oxidative stress, glutarylcarnitine, (C5-DC), demonstrated association with cGVHD. In contrast to aGVHD, patients with cGVHD had higher levels of C5-DC (OR = 4.97, 95% CI = (1.49, 21.99), p-value = 0.017). (Fig 3I–3K).

Compared to younger patients, baseline levels of various medium- and long-chain acylcarnitines [42] (p<0.05) were higher in older HCT patients. (Fig 4 and metabolomics S2 File).

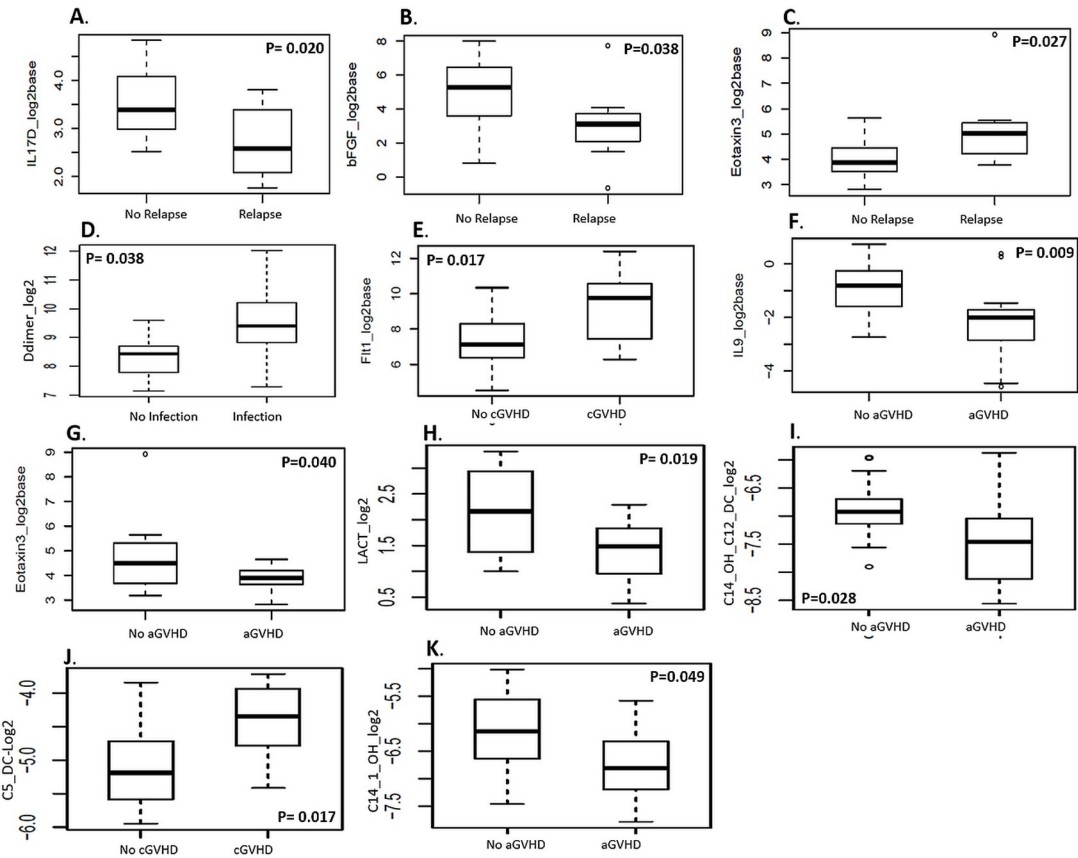

**Fig 3. Association between baseline biomarker levels and clinical outcome.** Data are presented as boxplots for immune/inflammatory response and metabolomic biomarkers that demonstrated significant association with post-HCT aGVHD, cGVHD, relapse or infection at baseline. Effect sizes of each biomarker for clinical outcomes (aGVHD, cGVHD, relapse and infection) were assessed using logistic regression models. Odds ratios (OR), score test p-values, and 95% CI were reported for the logistic regression models assessing biomarker effects on clinical outcomes.

## Discussion

Age related inflammation, also termed "inflammaging" is related to the activation/dysregulation of both innate and adaptive immune systems and is considered a significant risk factor in many age-related diseases [13, 16]. The exact mechanism of inflammaging and its contribution to adverse health outcomes are mostly unknown [16]. However, numerous studies have shown that several pro inflammatory cytokines, including IL-6 and TNFα increase with age in healthy individuals and in the absence of infection [43–45].

Of the 61 immune and inflammatory response related biomarkers we investigated, 17 were increased in older patients and only one marker (IL-23) was increased in younger patients pre-HCT compared to older patients. Unsurprisingly, most of the biomarkers elevated in elderly patients were inflammatory cytokines, such as IL16, IL17, MIP1a, TNF-α, and TNF receptors TNFRI and TNFRII. In addition, TNF-α an essential signaling protein in the innate and adaptive immune systems, is considered a biomarker of HCT treatment toxicity and a key cytokine in the effector phase of GVHD. TNF inhibitors have shown efficacy in clinical and experimental models of GVHD [46, 47]. Surprisingly, IL-23 levels were increased in younger patients compared to older patients. The differing expression levels of IL-23 and IL-17 were unexpected, since IL-23 and IL-17 are closely intertwined. IL-23 is known to drive promotion

| Biomarker | Median(Range) of Younger | Median(Range) of Older |
|---|---|---|
| C18:2-OH | 0.005 (0.001 - 0.01) | 0.008 (0.004 - 0.012) |
| C4-DC/Ci4-DC | 0.026 (0.01 - 0.036) | 0.031 (0.014 - 0.08) |
| C8:1-OH/C6:1-DC | 0.022 (0.011 - 0.038) | 0.03 (0.018 - 0.082) |
| C12-OH/C10-DC | 0.004 (0.002 - 0.013) | 0.008 (0.005 - 0.024) |
| C20-OH/C18-DC | 0.006 (0.003 - 0.011) | 0.009 (0.004 - 0.019) |
| C14:1-OH | 0.009 (0.005 - 0.018) | 0.015 (0.006 - 0.031) |
| C16:1 | 0.016 (0.004 - 0.045) | 0.028 (0 - 0.116) |
| C10-OH/C8-DC | 0.025 (0.007 - 0.065) | 0.043 (0.019 - 0.071) |
| C18-OH/C16-DC | 0.005 (0 - 0.01) | 0.007 (0.002 - 0.014) |
| C14-OH/C12-DC | 0.005 (0 - 0.016) | 0.008 (0 - 0.017) |
| C8:1-DC | 0.024 (0.013 - 0.045) | 0.033 (0.013 - 0.054) |
| C14:1 | 0.035 (0.02 - 0.175) | 0.07 (0.025 - 0.175) |
| C8:1 | 0.153 (0.041 - 0.329) | 0.237 (0.108 - 0.547) |
| C14:2 | 0.019 (0.01 - 0.089) | 0.04 (0.009 - 0.099) |
| C16-OH/C14-DC | 0.004 (0.002 - 0.008) | 0.006 (0.003 - 0.01) |
| C8 | 0.055 (0.025 - 0.158) | 0.092 (0.028 - 0.352) |
| C10:3 | 0.044 (0.015 - 0.111) | 0.054 (0.017 - 0.152) |
| C18:2 | 0.066 (0.018 - 0.257) | 0.124 (0.007 - 0.889) |
| C18:1-DC | 0.006 (0.001 - 0.014) | 0.009 (0.004 - 0.021) |
| C18:1-OH/C16:1-DC | 0.005 (0.003 - 0.012) | 0.007 (0 - 0.021) |

**Fig 4. Acylcarnitines metabolomics profiling differential expression in younger vs. older Patients ($\leq$30 years vs. $\geq$ 55 years old).** The Wilcoxon rank-sum test was used to compare metabolomic marker levels of baseline in different age groups (($\leq$30 years vs. $\geq$ 55 years old). Compared to younger patients, baseline levels of various medium- and long-chain acylcarnitines were higher in older patients.

of T helper type 17 (Th17) cells that secrete IL-17 [48]. This signaling pathway is critical in autoimmune conditions such as psoriasis and rheumatoid arthritis, and therapeutics targeting IL-17 or IL-23 are under clinical investigation [49]. In addition, ustekinumab, an antibody targeting IL-23 was reported to be effective in glucocorticoid-refractory aGVHD [50]. Other potential targets for aGVHD treatment involving the IL-23/IL-17 pathway, include the bromodomain and extra-terminal domain (BET) proteins. *In vitro* and *in vivo* assays [51], have demonstrated the potent anti-inflammatory effects of BET inhibition, and its effect on impacting IL-23R/IL-17 immune axis (decreasing the expression levels of IL23-R and IL17). BET inhibitors (Plexxikon-51107 and -2853 (PLX51107 and PLX2853)) have demonstrated that they can significantly improves survival and reduces aGVHD progression. PLX51107 will be studied in a phase Ib/II clinical trial for its effects on in treating Steroid-Refractory aGVHD (NCT04910152) [51].

Baseline levels of three markers (CRP, SAA and IL13) showed significant association with survival. Basal levels were lower in all the three markers in HCT patients with longer OS (OS$\geq$ 3 years). IL-13 is involved in Th2 inflammation and higher pre transplant levels of IL13 have been reported as a strong predictor of developing aGVHD [52]. IL13 and its receptors have been evaluated as a possible therapeutic target in different diseases including various types of cancer [52–55].

Clinically, CRP is frequently examined broad-scale in a variety of disease states. CRP, C-reactive protein, is often measured as a broad-scale marker of inflammation in infection and auto-immune/rheumatologic conditions [56]. SAA or serum-amyloid A is a group of apolipoproteins associated with high density lipoprotein (HDL) that is both expressed constitutively and in response to inflammation [57]. Additionally, SAA has been implicated in the suppressive effects of tumor cells on immune cells in both melanoma and glioblastoma models [58, 59].

Similar to CRP, D-dimer is an established biomarker of a number of disease states. D-dimer, a product of blood clot degradation, is most often measured when there is suspicion of pulmonary embolism or deep-vein thrombosis, but it can also be elevated in infection, inflammation, pregnancy, trauma, and malignancy [60]. The association between elevated D-dimer and infection post-HCT observed here might suggest a chronic inflammatory state, which combined with the immunosuppression after bone marrow transplant might lead to increased inability to fight off infection. CRP, D-dimer and SAA are notable for being highly upregulated during the "acute phase response"–the body's physiological reaction to stresses such as infection, inflammation, and trauma [61].

Of the 65 metabolomic biomarkers analyzed, we noted that lower mean levels of lactate were associated with aGVHD. A growing body of clinical evidence indicts dysregulation of lactate–pyruvate fuel metabolism in poor clinical outcomes after HCT [62, 63]. Lactate is the product of glycolysis and substrate for mitochondrial respiration. It has a variety of roles, including its importance as an energy source, a precursor for gluconeogenesis, and an essential signaling molecule [64]. Critically, T cells rely upon glycolysis and lactate to perform their effector functions. In fact, 13C-pyruvate MRI has been used to detect aGVHD in a mouse model of HCT. In that study, plasma levels of lactate were significantly elevated at day 7 post-HCT after allogenic but not syngeneic transplants [65]. This timing of lactate elevation could explain why we observed that a lower baseline level of lactate was associated with aGVHD.

Acylcarnitines also play an important role in energy metabolism by participating in the transfer of fatty acids into mitochondrial membrane [42, 66]. Acylcarnitines, esters of L-carnitines and fatty acids, are important intermediates in metabolism. They are responsible for transporting long chain fatty acids across the mitochondria for beta-oxidation. They have been used as biomarkers for inherited metabolic disorders since dysfunction in fatty acid or amino acid metabolism can lead to changes in plasma acylcarnitine concentration [67]. Our previous studies have shown that long-chain acylcarnitines, and carnitine esters of dicarboxylic acids are associated with cardiovascular disease, including heart failure [68–70]. Acylcarnitines can mediate inflammation [71], and have been proposed as biomarkers for hepatocellular carcinoma and liver dysfunction since the liver is the most active organ for acylcarnitine synthesis and metabolism [72].

Our results suggest that baseline levels of medium- and long-chain AC markers were higher in older HCT patients compared to younger HCT patients. In patients with aGVHD compared to patients who did not develop aGVHD, the baseline levels of certain long-chain acylcarnitines were lower. Baseline levels of the short chain AC marker, glutaryly carnitine, C5- DC [42], a marker of omega oxidation and oxidative stress, demonstrated association with cGVHD: patients with cGVHD had higher levels of C5-DC. Accumulation of medium- and long-chain acylcarnitines in older patients suggests incomplete beta-oxidation of fatty acids, resulting in lower flux of fatty hydrocarbon toward acetylcarnitine (AC C2) and, hence, acetyl CoA. To maintain overall energy balance, glycolytic pyruvate metabolism might shift away from lactic acid and toward production of acetyl CoA and Krebs cycling [73, 74].

Our data show a significant decrease in lactate with aGVHD, as they also do decreases in various ACs. This could potentially signify a pathway emphasis towards Krebs (and oxidative phosphorylation) and away from lactate or fatty acid synthesis, or alternatively a drive in the

opposite direction through pyruvate towards gluconeogenesis. In order to have the data and conclusion, to support such claims further in-depth studies are needed.

Strengths of this exploratory study include the large number of biomarkers (>100 total) analyzed from patient samples, the long-term survival data, and similar pre-HCT and peri-HCT conditioning treatment. The major limitation of this study is the small cohort size. The small sample size and the large number of the markers will not allow for the adjusted p-value analysis. Other limitations include disparate cohort sizes, lack of matched healthy control data and reliance on single institution data. In addition, we have only measured the baseline levels of the markers, which did not evaluate the longitudinal changes of these markers (including known GVHD markers REG3 and ST2, that did not show significant association in our baseline analysis). As previously stated this retrospective study was designed with an exploratory and hypothesis-generating approach to comprehensively characterize immune, inflammatory, and metabolomic biomarkers. Thus, biomarkers highlighted here should be subjected to further analyses.

In conclusion, we have identified several inflammatory and metabolomic biomarkers that are differentiate young vs old HCT patients and are associated with survival and clinical outcomes in HCT. Many of these biomarkers have been previously associated with malignancy or immune responses. Given the generally poor outcomes for older patients with AML, additional investigation is warranted. A combination of biomarkers could guide interventions, and personalized immune-modulatory therapeutics post-HCT could help to improve clinical outcomes in AML.

## Supporting information

**S1 File.**
(ZIP)

**S2 File.**
(PDF)

## Acknowledgments

We a like to thank Huaxia Cui and Tabitha George for their assistance in with metabolomics assay measurements.

## Author Contributions

**Conceptualization:** Sharareh Siamakpour-Reihani, Anthony D. Sung.

**Data curation:** Felicia Cao.

**Formal analysis:** Jing Lyu, Yi Ren, Jichun Xie, Mark D. Starr.

**Funding acquisition:** Anthony D. Sung.

**Investigation:** Sharareh Siamakpour-Reihani, Nelson J. Chao, Anthony D. Sung.

**Methodology:** Sharareh Siamakpour-Reihani, Andrew B. Nixon, Amy T. Bush, Mark D. Starr, James R. Bain, Michael J. Muehlbauer, Olga Ilkayeva, Virginia Byers Kraus, Janet L. Huebner, Anthony D. Sung.

**Project administration:** Sharareh Siamakpour-Reihani.

**Resources:** Amy T. Bush, Nelson J. Chao, Anthony D. Sung.

**Supervision:** Sharareh Siamakpour-Reihani, Andrew B. Nixon, Jichun Xie, James R. Bain, Virginia Byers Kraus, Anthony D. Sung.

**Writing – original draft:** Sharareh Siamakpour-Reihani, Felicia Cao.

**Writing – review & editing:** Sharareh Siamakpour-Reihani, Felicia Cao, Jing Lyu, Yi Ren, Andrew B. Nixon, Jichun Xie, Amy T. Bush, Michael J. Muehlbauer, Olga Ilkayeva, Virginia Byers Kraus, Janet L. Huebner, Nelson J. Chao, Anthony D. Sung.

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
