## [Decision Letter · Decision Letter 0]

10 Jan 2022

PONE-D-21-39481Evaluating immune response and metabolic related biomarkers pre-allogenic hematopoietic stem cell transplant in acute myeloid leukemiaPLOS ONE

Dear Dr. Sung,

Thank you for submitting your manuscript to PLOS ONE. After careful consideration, we feel that it has merit but does not fully meet PLOS ONE’s publication criteria as it currently stands. Therefore, we invite you to submit a revised version of the manuscript that addresses the points raised during the review process by both Reviewers, experts in the field.

We look forward to receiving your revised manuscript.

Kind regards,

Francesco Bertolini, MD, PhD

Academic Editor

PLOS ONE

Journal Requirements:

"This research was in part supported, in part, by the American Society of Hematology (ASH) Scholar Award and the NIH/National Institute on Aging 1R21AG066388-01 award"

"YES - Specify the role(s) played

A.D.S.:

American Society of Hematology (ASH) 

https://www.hematology.org/

Scholar Award and the NIH/National Institute on Aging 1R21AG066388-01 award.

" ext-link-type="uri" xlink:type="simple">https://reporter.nih.gov/search/5NyHmfz0skuWJC2JAuThRw/project-details/9980757"

Reviewers' comments:

Reviewer's Responses to Questions

**Comments to the Author**

1. Is the manuscript technically sound, and do the data support the conclusions?

Reviewer #1: Yes

Reviewer #2: Partly

2. Has the statistical analysis been performed appropriately and rigorously? 

Reviewer #1: Yes

Reviewer #2: Yes

3. Have the authors made all data underlying the findings in their manuscript fully available?

Reviewer #1: Yes

Reviewer #2: Yes

4. Is the manuscript presented in an intelligible fashion and written in standard English?

Reviewer #1: Yes

Reviewer #2: Yes

5. Review Comments to the Author

Reviewer #1: Summary: In the present study, the authors evaluated immune/inflammatory responses and metabolic biomarkers associated with overall survival (OS) in acute myeloid leukemia (AML) after hematopoietic stem cell transplantation (HCT). Using plasma from 34 AML patients after HCT, they analyzed 65 blood-based metabolomics markers and 61 immune/inflammatory related biomarkers, comparing long-term OS (1 year) with short-term OS (3 years), or comparing younger (30 years) with older (55 years) patients. They found several markers that were elevated or reduced in older versus younger patients, as well as many immune/inflammatory markers associated with outcomes such as graft-versus-host disease (GVHD), infection, or relapse. For metabolic markers, they found higher levels of medium- and long-chain acylcarnitines in older patients with GVHD. Further studies would be required to assess the use of these biomarkers in disease prognosis and therapy. Comments for improvement are listed below:

Comments:

1. Overall survival (OS) is spelled out twice in the abstract.

2. Abstract Line 32: “This retrospective study was designed with an exploratory and hypothesis generating….” Introduction Line 113-114, “We hypothesized that metabolomics and blood-based biomarkers have the potential to be used as predictive biomarkers for AML patients who receive HCT treatment.” Please clearly state the hypothesis within the abstract.

3. Please spell out all abbreviations upon first usage. For example, MMP-3 is spelled out on line 152, but first used on line 99.

4. Methods: For biomarkers of inflammation and aging, please provide more details regarding how the 54-plex plates were read. What equipment did you use? Were serial dilutions performed?

5. The metabolomics data are presented haphazardly in the Results section. They are hard to follow. Revision is recommended.

6. Discussion: A recent article (PMID: 34722316) regarding BET inhibition in GVHD after HCT should be discussed in relation to IL-23/IL-17.

7. Discussion Lines 341-346 are not necessary and do not have relevance to the current manuscript.

8. Do the authors speculate that increased acylcarnitines in older patients correlates with enhanced oxidative phosphorylation? In what cell type? This would be supported by the lower mean levels of lactate. Please clarify within the text.

9. Figure 1, panel labels are recommended (e.g. A, B, C, D). These should be updated within the text. Please use consistent font and font sizes for all graph labels.

10. Figure legends: Please indicate the meaning of * within the figures.

11. There is a lot of supporting information that is not described or called out within the manuscript text.

Reviewer #2: Siamakpour-Reihani et al investigate the expression of a large number of immune/inflammatory and metabolomic biomarkers in plasma samples from patients with acute myeloid leukemia (AML) before hematopoietic stem cell transplantation (HCT) and explore associations with disease outcome (overall survival-OS), young (≤30years) or old age (≥ 60 years) and other clinical outcomes (chronic graft versus host disease-cGVHD, acute graft versus host disease-aGVHD, infection and relapse). They found that several biomarkers were elevated in older patients compared to younger patients, whereas only IL23 was upregulated in younger patients. Three markers (IL13, SAA, CRP) were lower in patients with OS ≥ 3 years, whereas IL-9, Eotaxin-3 was associated with aGVHD, Flt-1 with cGVHD, D-dimer with infection, or IL-17D, bFGF, and Eotaxin-3 with relapse. Higher baseline levels of medium- and long-chain acylcarnitines (AC) were found in older patients, whereas lactate and long-chain AC were associated with aGVHD, and medium-chain AC with cGVHD. These findings are interesting, and the manuscript is well-presented; however, the following issues need attention.

1. Sex and age matched healthy controls are missing. Thus, it is not clear whether the differential expression of immune/inflammatory and metabolomic biomarkers in plasma samples from AML patients owes to disease, age or comorbidities related to elder.

2. The characteristics of the patients in Table 1, should be presented in more detail. All the parameters examined (cGVHD, aGVHD, infection, relapse, deaths, etc), as well as disease comorbidities (such as thrombosis, cardiac disease, etc) must be presented.

3. In this context, it should be excluded that the factors that are differentially expressed in the various subsets of AML patients, do not owe their differential expression to an infection or comorbidity.

4. Metabolomic analysis was performed in retrospectively collected samples. Were the samples collected appropriately for this kind of analysis? The samples were collected at similar period of the day, were the patients fasting? All this information needs to be included in the manuscript (if they were not performed, along with comment #1, they should be discussed).

5. It is not reported why the authors chose these age limits (e.g., why young were considered those under 30 years and not those under 32 or 35 years). It was based on statistical analysis, or it was an arbitrary selection? This must be clearly stated.

6. PLOS authors have the option to publish the peer review history of their article (what does this mean?). If published, this will include your full peer review and any attached files.

Reviewer #1: No

Reviewer #2: No

---

## [Author Response · Author response to Decision Letter 0]

17 Mar 2022

PONE-D-21-39481

03/04/2022

Rebuttal letter

We are greatly appreciative for this careful and detailed review. Please see a point-by-point response to the reviewers’ comments. We have revised the paper accordingly. 

Journal Requirements:

2. and 

Response: Written, witnessed informed consent was obtained for sample collection for future research purposes, as well as use of associated clinical data. De-identified samples and de-identified clinical data were accessed under a separate, IRB-approved protocol; because deidentified samples and data were used, and patients already consented to future use of samples for research purposes, the IRB waived the requirement for consent for this use protocol. (IRB #s are: Pro00006268 and Pro00100650).: 

"This research was in part supported, in part, by the American Society of Hematology (ASH) Scholar Award and the NIH/National Institute on Aging 1R21AG066388-01 award"

"YES - Specify the role(s) played

A.D.S.:

American Society of Hematology (ASH) 

https://www.hematology.org/

Scholar Award and the NIH/National Institute on Aging 1R21AG066388-01 award.

https://reporter.nih.gov/search/5NyHmfz0skuWJC2JAuThRw/project-details/9980757"

Response: We have removed funding-related text from the manuscript. In the cover letter we have requested the following funding information (* is a new request):

We also would like to update our Funding Statement to include the following information (per your request this section has bene removed from the acknowledge section): 

1. American Society of Hematology (ASH), (PI: Anthony D Sung)

https://www.hematology.org/

2. NIH/National Institute on Aging 1R21AG066388-01 award (PI: Anthony D Sung).

https://reporter.nih.gov/search/5NyHmfz0skuWJC2JAuThRw/project-details/9980757

3. NIH/National Institute on Aging Duke Pepper Older Americans Independence Center 

P30 AG028716, (PI: Schmader, Mini #6, PI of Mini: Anthony D Sung) 

https://reporter.nih.gov/search/KsvVX_8rwUKk6CXVf2k0vg/project-details/9971412

Response: The minimal data set underlying the results described in our manuscript have been uploaded as supplement/ supporting document. We have also added this statement to the cover letter: “This revised manuscript includes the revisions and edits requested by reviewers. In addition, for the re-submission of our revised manuscript, we have uploaded our study’s minimal underlying data set as Supporting Information files.” 

Reviewers' comments:

Reviewer's Responses to Questions

Comments to the Author

Reviewer #1: Comments:

1. Overall survival (OS) is spelled out twice in the abstract.

Response: This has been corrected 

2. Abstract Line 32: “This retrospective study was designed with an exploratory and hypothesis generating….” Introduction Line 113-114, “We hypothesized that metabolomics and blood-based biomarkers have the potential to be used as predictive biomarkers for AML patients who receive HCT treatment.” Please clearly state the hypothesis within the abstract.

Response: Revisions have been added 

3. Please spell out all abbreviations upon first usage. For example, MMP-3 is spelled out on line 152, but first used on line 99. 

Response: Revisions have been added

4. Methods: For biomarkers of inflammation and aging, please provide more details regarding how the 54-plex plates were read. What equipment did you use? Were serial dilutions performed?

Response: Revisions have been added to the methods sub- section for “Biomarkers of inflammation and aging”

5. The metabolomics data are presented haphazardly in the Results section. They are hard to follow. Revision is recommended.

Response: The section has been revised in results section.

6. Discussion: A recent article (PMID: 34722316) regarding BET inhibition in GVHD after HCT should be discussed in relation to IL-23/IL-17.

Response: This has been added to the discussion section

7. Discussion Lines 341-346 are not necessary and do not have relevance to the current manuscript.

Response: Section removed 

8. Do the authors speculate that increased acylcarnitines in older patients correlates with enhanced oxidative phosphorylation? In what cell type? This would be supported by the lower mean levels of lactate. Please clarify within the text.

Response: We show no significant difference with age for lactate, so would not appear to support an association with the increase in long/median-chain AC with age. Our data do however show a significant decrease in lactate with aGVHD, as they also do decreases in various ACs. This could potentially signify a pathway emphasis towards Krebs (and oxidative phosphorylation) and away from lactate or fatty acid synthesis, or alternatively a drive in the opposite direction through pyruvate towards gluconeogenesis. In order to have the data and conclusion, we would need a more intensive study to support such claims (this would be out of the scope for this paper).

A paragraph also has been added with references to the discussion related to this comment.

9. Figure 1, panel labels are recommended (e.g. A, B, C, D). These should be updated within the text. Please use consistent font and font sizes for all graph labels.

Response: The figure has been revised and updated within the text 

10. Figure legends: Please indicate the meaning of * within the figures. 

Response: Please note there is no * sign in the figures. The only * signs are for table 2 with the explanation of: “* are separate ELISAs not part of the 54-plex. ** Higher baseline levels in younger patients”.

However, in the figures there are dots/points (which might have been mistaken for *), so we have added the following to Fig 1 legend: data is presented as a boxplot. Boxes represent 25th (Q1) and 75th (Q3) percentiles; Horizontal lines indicate the medians; Upper whiskers indicate maximum; Lower whiskers indicate minimum. Points indicate any observations outside the whiskers.”

11. There is a lot of supporting information that is not described or called out within the manuscript text.

Response: We have added more data and supporting information, so we hope we have responded to the comment to an extent (since the comment is general, we did not know if adding specific information was intended). 

Reviewer #2: Siamakpour-Reihani et al investigate the expression of a large number of immune/inflammatory and metabolomic biomarkers in plasma samples from patients with acute myeloid leukemia (AML) before hematopoietic stem cell transplantation (HCT) and explore associations with disease outcome (overall survival-OS), young (≤30years) or old age (≥ 60 years) and other clinical outcomes (chronic graft versus host disease-cGVHD, acute graft versus host disease-aGVHD, infection and relapse). They found that several biomarkers were elevated in older patients compared to younger patients, whereas only IL23 was upregulated in younger patients. Three markers (IL13, SAA, CRP) were lower in patients with OS ≥ 3 years, whereas IL-9, Eotaxin-3 was associated with aGVHD, Flt-1 with cGVHD, D-dimer with infection, or IL-17D, bFGF, and Eotaxin-3 with relapse. Higher baseline levels of medium- and long-chain acylcarnitines (AC) were found in older patients, whereas lactate and long-chain AC were associated with aGVHD, and medium-chain AC with cGVHD. These findings are interesting, and the manuscript is well-presented; however, the following issues need attention.

1. Sex and age matched healthy controls are missing. Thus, it is not clear whether the differential expression of immune/inflammatory and metabolomic biomarkers in plasma samples from AML patients owes to disease, age or comorbidities related to elder.

Response: lack of matched healthy controls is a limitation of our study, we have add this to our “limitation section of the manuscript”. In this retrospective study the comparison is only made between the blood based biomarkers in different groups of AML patients based on age and survival. Also, we did not have access to matched healthy control samples for this retrospective study.

2. The characteristics of the patients in Table 1, should be presented in more detail. All the parameters examined (cGVHD, aGVHD, infection, relapse, deaths, etc), as well as disease comorbidities (such as thrombosis, cardiac disease, etc) must be presented.

3. In this context, it should be excluded that the factors that are differentially expressed in the various subsets of AML patients, do not owe their differential expression to an infection or comorbidity.

Response (#2): Table 1 have been updated to present the parameters and conditions examined.

Response (#3): Based on the updated Table 1, we have not seen balance issue between age groups in our outcomes. The p-values are not significant for differences between groups for infection or other comorbidities we studied.

The only two significant p-values for difference between the two groups in Table 1 are: for the age (which was by design), and for Conditioning (which is expected since the conditioning for younger HCT patients will be different 

4. Metabolomic analysis was performed in retrospectively collected samples. Were the samples collected appropriately for this kind of analysis? The samples were collected at similar period of the day, were the patients fasting? All this information needs to be included in the manuscript (if they were not performed, along with comment #1, they should be discussed).

Response: All samples were appropriate for the metabolomics assays as the DPMI core is highly specialized in these studies. 

The blood samples have been collected as part of the ABMT Biorepository protocol and we used them retrospectively. However, the samples are not collected at the same time of the day. As part of our IRB approved protocol the biorepository samples have been collected during routine appointments pre and post HCT at routine lab visits. Thus, they are not all collected at the same time. The patients are not required to be fasting unless requested by the physician. 

5. It is not reported why the authors chose these age limits (e.g., why young were considered those under 30 years and not those under 32 or 35 years). It was based on statistical analysis, or it was an arbitrary selection? This must be clearly stated. 

Response: The information was added to the methods section: The age cut-offs were arbitrary selection and mostly based on what we had samples for. Though there is support that AML in patients 55years behaves differently than AML in patients 55 years (Appelbaum, Gundacker et al. 2006). 

Appelbaum, F. R., H. Gundacker, D. R. Head, M. L. Slovak, C. L. Willman, J. E. Godwin, J. E. Anderson and S. H. Petersdorf (2006). "Age and acute myeloid leukemia." Blood 107(9): 3481-3485.

---

## [Decision Letter · Decision Letter 1]

3 Apr 2022

PONE-D-21-39481R1Evaluating immune response and metabolic related biomarkers pre-allogenic hematopoietic stem cell transplant in acute myeloid leukemiaPLOS ONE

Dear Dr. Sung,

Thank you for submitting your manuscript to PLOS ONE. After careful consideration, we feel that it has merit but does not fully meet PLOS ONE’s publication criteria as it currently stands. Therefore, we invite you to submit a revised version of the manuscript that addresses the points raised during the review process by Reviewer #1.

We look forward to receiving your revised manuscript.

Kind regards,

Francesco Bertolini, MD, PhD

Academic Editor

PLOS ONE

Journal Requirements:

Reviewers' comments:

Reviewer's Responses to Questions

**Comments to the Author**

1. If the authors have adequately addressed your comments raised in a previous round of review and you feel that this manuscript is now acceptable for publication, you may indicate that here to bypass the “Comments to the Author” section, enter your conflict of interest statement in the “Confidential to Editor” section, and submit your "Accept" recommendation.

Reviewer #1: All comments have been addressed

Reviewer #2: All comments have been addressed

2. Is the manuscript technically sound, and do the data support the conclusions?

Reviewer #1: Yes

Reviewer #2: Yes

3. Has the statistical analysis been performed appropriately and rigorously? 

Reviewer #1: No

Reviewer #2: Yes

4. Have the authors made all data underlying the findings in their manuscript fully available?

Reviewer #1: Yes

Reviewer #2: Yes

5. Is the manuscript presented in an intelligible fashion and written in standard English?

Reviewer #1: Yes

Reviewer #2: Yes

6. Review Comments to the Author

Reviewer #1: Summary: In the present study, the authors evaluated immune/inflammatory responses and metabolic biomarkers associated with overall survival (OS) in acute myeloid leukemia (AML) after hematopoietic stem cell transplantation (HCT). Using plasma from 34 AML patients after HCT, they analyzed 65 blood-based metabolomics markers and 61 immune/inflammatory related biomarkers, comparing long-term OS (1 year) with short-term OS (3 years), or comparing younger (30 years) with older (55 years) patients. They found several markers that were elevated or reduced in older versus younger patients, as well as many immune/inflammatory markers associated with outcomes such as graft-versus-host disease (GVHD), infection, or relapse. For metabolic markers, they found higher levels of medium- and long-chain acylcarnitines in older patients with GVHD. Further studies would be required to assess the use of these biomarkers in disease prognosis and therapy. Comments for improvement are listed below:

Comments:

1. Please provide statistics within each panel of every figure.

2. Supplemental data are still not adequately called out within the text.

Reviewer #2: The authors addressed all issues and the manuscript is appropriate for publication.

The only correction that is needed in the new version is the deletion of the phrase in lines 350-351 because it is a repetition of the next phrase.

7. PLOS authors have the option to publish the peer review history of their article (what does this mean?). If published, this will include your full peer review and any attached files.

Reviewer #1: No

Reviewer #2: No

---

## [Author Response · Author response to Decision Letter 1]

12 Apr 2022

04-11-2022

Rebuttal letter: 

Thank you for your time and comments. We are greatly appreciative for this second review on the revisions previously made. Please see the response to the additional comments and minor changes requested (blue color font). We have revised the paper accordingly. 

 PONE-D-21-39481R1

Evaluating immune response and metabolic related biomarkers pre-allogenic hematopoietic stem cell transplant in acute myeloid leukemia

PLOS ONE

Response: This had been done

Response: This had been done

• Response: This had been done

Reviewers' comments:

Reviewer's Responses to Questions

Comments to the Author

1. If the authors have adequately addressed your comments raised in a previous round of review and you feel that this manuscript is now acceptable for publication, you may indicate that here to bypass the “Comments to the Author” section, enter your conflict of interest statement in the “Confidential to Editor” section, and submit your "Accept" recommendation.

Reviewer #1: All comments have been addressed

Reviewer #2: All comments have been addressed

2. Is the manuscript technically sound, and do the data support the conclusions?

Reviewer #1: Yes

Reviewer #2: Yes

3. Has the statistical analysis been performed appropriately and rigorously?

Reviewer #1: No

Reviewer #2: Yes

4. Have the authors made all data underlying the findings in their manuscript fully available?

Reviewer #1: Yes

Reviewer #2: Yes

5. Is the manuscript presented in an intelligible fashion and written in standard English?

Reviewer #1: Yes

Reviewer #2: Yes

6. Review Comments to the Author

Reviewer #1: Summary: In the present study, the authors evaluated immune/inflammatory responses and metabolic biomarkers associated with overall survival (OS) in acute myeloid leukemia (AML) after hematopoietic stem cell transplantation (HCT). Using plasma from 34 AML patients after HCT, they analyzed 65 blood-based metabolomics markers and 61 immune/inflammatory related biomarkers, comparing long-term OS (1 year) with short-term OS (3 years), or comparing younger (30 years) with older (55 years) patients. They found several markers that were elevated or reduced in older versus younger patients, as well as many immune/inflammatory markers associated with outcomes such as graft-versus-host disease (GVHD), infection, or relapse. For metabolic markers, they found higher levels of medium- and long-chain acylcarnitines in older patients with GVHD. Further studies would be required to assess the use of these biomarkers in disease prognosis and therapy. Comments for improvement are listed below:

Comments:

1. Please provide statistics within each panel of every figure.

Response: Thank you for your suggestion. We have added information requested to the figures. Figures 1-3 have been revised. Some additional statistical data has also been added to the figure legends and the main text in the revised manuscript. 

2. Supplemental data are still not adequately called out within the text.

Response: Information has been added to the manuscript (Lanes 198 and 281)

Reviewer #2: The authors addressed all issues and the manuscript is appropriate for publication.

The only correction that is needed in the new version is the deletion of the phrase in lines 350-351 because it is a repetition of the next phrase.

Response: The phrase has been deleted / revised

---

## [Decision Letter · Decision Letter 2]

12 May 2022

Evaluating immune response and metabolic related biomarkers pre-allogenic hematopoietic stem cell transplant in acute myeloid leukemia

PONE-D-21-39481R2

Dear Dr. Sung,

We’re pleased to inform you that your manuscript has been judged scientifically suitable for publication and will be formally accepted for publication once it meets all outstanding technical requirements.

Kind regards,

Francesco Bertolini, MD, PhD

Academic Editor

PLOS ONE

Additional Editor Comments (optional):

Reviewers' comments:

Reviewer's Responses to Questions

**Comments to the Author**

1. If the authors have adequately addressed your comments raised in a previous round of review and you feel that this manuscript is now acceptable for publication, you may indicate that here to bypass the “Comments to the Author” section, enter your conflict of interest statement in the “Confidential to Editor” section, and submit your "Accept" recommendation.

Reviewer #1: All comments have been addressed

Reviewer #2: All comments have been addressed

2. Is the manuscript technically sound, and do the data support the conclusions?

Reviewer #1: Yes

Reviewer #2: Yes

3. Has the statistical analysis been performed appropriately and rigorously? 

Reviewer #1: Yes

Reviewer #2: Yes

4. Have the authors made all data underlying the findings in their manuscript fully available?

Reviewer #1: Yes

Reviewer #2: Yes

5. Is the manuscript presented in an intelligible fashion and written in standard English?

Reviewer #1: Yes

Reviewer #2: No

6. Review Comments to the Author

Reviewer #1: The authors have adequately addressed my concerns, specifically regarding statistical analysis and references the supplemental materials throughout the text.

Reviewer #2: (No Response)

7. PLOS authors have the option to publish the peer review history of their article (what does this mean?). If published, this will include your full peer review and any attached files.

Reviewer #1: No

Reviewer #2: No

---

## [Editor Report · Acceptance letter]

1 Jun 2022

PONE-D-21-39481R2 

Evaluating immune response and metabolic related biomarkers pre-allogenic hematopoietic stem cell transplant in acute myeloid leukemia 

Dear Dr. Sung:

I'm pleased to inform you that your manuscript has been deemed suitable for publication in PLOS ONE. Congratulations! Your manuscript is now with our production department. 

Kind regards, 

on behalf of

Dr. Francesco Bertolini 

Academic Editor

PLOS ONE